# A New Pathogenic Variant in *POU3F4* Causing Deafness Due to an Incomplete Partition of the Cochlea Paved the Way for Innovative Surgery

**DOI:** 10.3390/genes12050613

**Published:** 2021-04-21

**Authors:** Ahmet M. Tekin, Marco Matulic, Wim Wuyts, Masoud Zoka Assadi, Griet Mertens, Vincent van Rompaey, Yongxin Li, Paul van de Heyning, Vedat Topsakal

**Affiliations:** 1Department of Otorhinolaryngology, Head and Neck Surgery, Brussels Health Campus, Vrije Universiteit Brussel, 1090 Brussels, Belgium; drtekinahmet@gmail.com (A.M.T.); marco.matulic@cascination.com (M.M.); 2Center of Medical Genetics, Faculty of Medicine and Health Sciences, University of Antwerp and Antwerp University Hospital, 2650 Antwerp, Belgium; wim.wuyts@uantwerpen.be; 3MED-EL Medical Electronics, 6020 Innsbruck, Austria; masoud.zoka@medel.com; 4Department of Otorhinolaryngology, Head and Neck Surgery, Antwerp University Hospital, 2650 Edegem, Belgium; griet.mertens@uza.be (G.M.); vincent.van.rompaey@uza.be (V.v.R.); paul.vandeheyning@uza.be (P.v.d.H.); 5Department of Translational Neurosciences, Faculty of Medicine and Health Sciences, University of Antwerp, 2610 Antwerp, Belgium; 6Department of Otolaryngology, Head and Neck Surgery, Capital Medical University, Beijing 100730, China; entlyx@sina.com; 7Department of Otorhinolaryngology, Head and Neck Surgery, University Hospital UZ Brussel, Brussels Health Campus, Vrije Universiteit Brussel, 1090 Brussels, Belgium

**Keywords:** DFNX2, *POU3F4*, IP-III anomaly, sensorineural hearing loss, image guided surgery, robotically assisted cochlear implantation surgery

## Abstract

Incomplete partition type III (IP-III) is a relatively rare inner ear malformation that has been associated with a *POU3F4* gene mutation. The IP-III anomaly is mainly characterized by incomplete separation of the modiolus of the cochlea from the internal auditory canal. We describe a 71-year-old woman with profound sensorineural hearing loss diagnosed with an IP-III of the cochlea that underwent cochlear implantation. Via targeted sequencing with a non-syndromic gene panel, we identified a heterozygous c.934G > C p. (Ala31Pro) pathogenic variant in the *POU3F4* gene that has not been reported previously. IP-III of the cochlea is challenging for cochlear implant surgery for two main reasons: liquor cerebrospinalis gusher and electrode misplacement. Surgically, it may be better to opt for a shorter array because it is less likely for misplacement with the electrode in a false route. Secondly, the surgeon has to consider the insertion angles of cochlear access very strictly to avoid misplacement along the inner ear canal. Genetic results in well describes genotype-phenotype correlations are a strong clinical tool and as in this case guided surgical planning and robotic execution.

## 1. Introduction

### 1.1. POU3F4 Gene

Congenital hearing loss is one of the most common inherited human pathologies that occur in 1–2 out of 1000 newborns [1]. Approximately half of all cases seen are inherited, and they constitute a recessive autosomal pattern (70–80%), dominant autosomal pattern (10–20%), and (1–5%) X-linked hearing losses [2]. It is estimated that 20% of congenital disorders are attributed to inner ear malformation (IEM) [3]. X-Linked deafness-2 (DFNX2) (OMIM# 304400), also known as conductive deafness with congenital stapes footplate fixation (DFN3) [4,5], is accompanied by IEM, constitutes approximately half of the X-linked non-syndromic hearing loss and is found on chromosome Xq21.1. It is the type of deafness caused by pathogenic variants in the *POU3F4* gene [6].

Douville et al. reported that Pou3f4 is expressed during embryonic development in the brain, neural tube, and otic vesicle in rats [7]. Frameshift truncation and extension mutations in the C-terminal of *POU3F4* are reported to cause cytoplasmic localization and then proteasomal degradation due to structural abnormalities, leading to non-syndromic hearing loss [8]. The *POU3F4* gene encodes a transcription factor that consists of two parts, the POU-specific domain, and homeodomain, belonging to the POU-domain family [9,10]. It is known that the interactions between mesenchymal cells and spiral ganglion neurons expressing Pou3f4 are important for proper axon bundling during development [11]. Pou3f4 mRNA is expressed in mesenchymal tissue near the stapes footplate in the dorsal aspect of the annular stapedial ligament surrounding the otic vesicle during development. However, it is not expressed in the developing malleus and incus [12]. In addition, in mice with Pou3f4 deficiency, it has been reported that changes in cochlear spiral ligament fibrocytes cause a decrease in endocochlear potential and loss of expression in Kir4.1 channel in strial intermediate cells, which are required for stria vascularis conduction. This suggests that Pou3f4 deficiency causes defects in both otic fibrocytes and stria vascularis at different developmental stages and with different pathophysiological mechanisms [13].

### 1.2. Molecular and Clinical Characterization of POU3F4 Pathogenic Variants (Incl. Surgical Challenges)

Studies of the *POU3F4* pathogenic variants have shown that hearing loss results from a functional deficiency of the POU3F4 protein [14] and various pathogenic hearing loss causing variants have been described in this gene, including intra-genic, partial, or complete deletions of the gene, and deletions, inversions, and duplications of the *POU3F4* genomic region that do not include the *POU3F4* coding sequence [15]. Despite this, DFNX2 occurrence due to *POU3F4* pathology is rare compared to other genetic hearing losses. The hearing loss in patients with different *POU3F4* pathogenic variant types vary greatly. Thus, conduction type, mixed or sensorineural hearing loss (SNHL) can be observed in patients with *POU3F4* pathogenic variants [16,17,18,19,20,21,22,23,24,25,26,27,28,29,30] (Table 1).

This hearing impairment, also known as DFNX2, was described by Nance in 1971 as X-linked deafness in men and characterized by profound mixed hearing loss, vestibular abnormalities, and congenital fixation at the stapes with perilymphatic gusher [43]. Later, Papadaki et al. reported two cases of X-linked deafness with stapes gusher in women with normal male relatives [44]. It is argued that female heterozygotes are similar to males but have milder audiological abnormalities and they generally have SNHL due to modiolar defect and conductive hearing loss due to stapes fixation [45]. Computed tomography (CT) studies of developmental anomalies in the inner ear and middle ear in DFNX2 patients have shown an abnormal expansion of the internal auditory canal (IAC) and abnormal communication between the IAC and the inner ear compartments [31,46,47]. Jackler et in 1987 [48], after classifying IEM according to embryogenesis, Sennaroğlu and Saatci described the IEM classification through established CT and magnetic resonance imaging (MRI) methods in 2002 that is still generally accepted today [49]. According to this classification, in Incomplete partition type I (IP-I) (cystic cochlea-vestibular malformation), the cochlea that looks similar to a cystic cavity is devoid of septum and modiol and is accompanied by an enlarged vestibule. IP-II (classical Mondini’s deformity) is characterized by cystic cochlear apex with normal basal turns, enlarged vestibule, and enlarged vestibular aqueduct (LVA). The radiological appearance of IP-III is characteristic, and it was first defined by Phelps [46], and this image was classified in IP-III category by Sennaroğlu [50] in 2006. IP-III is a relatively rare IEM that accounts for about 2% of all cochlear malformations [44] and has been associated with *POU3F4* in studies [37,40,42,51]. The IP-III anomaly is mainly characterized by incomplete separation of the modiolus of the cochlea from the IAC. The modiolus is absent but interscalar septa are present and the dimensions of the cochlea are generally normal. Considering a large number of young/new cochlear implantation (CI) surgeons beginning to practice CI, 3D segmentation of the anatomic structure before surgery can provide accurate identification of IEM types and helps the operating surgeon to better understand the deformity and reduce surgical risks. Figure 1 shows the 3D segmentation of normal anatomy and IEM [52].

### 1.3. Cochlear Implantation Surgery for Sensorineural Hearing Loss

CI is the most effective neural prosthesis in humans to rehabilitate profound SNHL. DFNX2 patients were classified in the IP-III group, which led to some difficulties in CI application in the cochlea that may occur during or after surgery. Although audiological outcomes CI in IP-III anomalies have been noted to be slightly inferior to a normal cochlea [34,53,54], CI is still a unique and effective treatment for severe to profound SNHL in IP-III patients [32,33,35,55,56,57]. We present the case of a 71-year-old woman with profound SNHL diagnosed with an IP-III of the cochlea that underwent robotically-assisted cochlear implantation surgery (RACIS). This clinical condition is very rare, and its anatomy is very challenging for CI surgeons for two main reasons: gusher and electrode misplacement [45,58]. The lack of separation will inevitably provoke a gusher upon opening of the inner ear. Generally, the gusher will stop after some time depending on compensating mechanisms where membranes will close [59]. Nevertheless, specific surgical techniques have been developed such as harvesting temporalis muscle to pack around the cochleostomy [58,59,60] and extended round window insertion in CI [57]. Additionally, specific electrodes have been developed to stop oozing of perilymph after insertion such as the cork electrode [61] and to prevent unwanted facial nerve stimulation such as the slim electrode [62]. Secondly, insertion of a CI array has an increased risk of being misplaced towards the IAC. Often surgeons will opt for a shorter array because it is less likely to take that false route [59]. However, the surgical skill for a correct insertion depends on the angles under which the array is inserted. Related to the insertion angles of inner ear access RACIS has demonstrated its precision and accuracy [63]. Some surgeons argue that insertion of the array could be performed under fluoroscopy [55,58]; however, this is not helpful for deciding how to insert. This technique merely provides immediate feedback on misplacement allowing instant correction. However, damage to the auditory nerve at the level of IAC after misplacement may impair the functioning of a correctly placed CI in the cochlea.

RACIS pursues a minimally invasive direct access towards a designated target instead of a larger surgical exposure. Surgical exposure requires the identification of landmarks to work safely [63], whereas RACIS uses data, sometimes beyond human perception, to warrant safety and accuracy. Since we know the importance of pre-operative planning, intra-operative imaging, and the use of landmarks in IP-III anomaly cases due to the malformed anatomical structures, our aim was to study whether RACIS would enable evaluation and confirmation of the optimal insertion angle.

## 2. Materials and Methods

### 2.1. The Patient

A 71-year-old woman with profound SNHL diagnosed with an IP-III was referred for diagnostic work-up and candidacy selection for cochlear implantation in her right ear.

### 2.2. Audiological Evaluation

Baseline hearing tests were performed before and after surgery as part of the standard cochlear implantation workup. They included standard pure-tone audiometry and speech audiometry in quiet, with and without hearing aids included, and performing according to ISO 8253–1 (2010) standards to obtain pure air tone and bone conduction thresholds. The hearing thresholds were determined using pulsed pure tones in the frequency range of 125 Hz to 8 kHz.

### 2.3. Molecular Analysis

Molecular analysis was performed in the index patient using standard techniques on DNA extracted from fresh blood. First, a gene panel consisting of 84 genes known to play a role in non-syndromic hearing loss was enriched with Haloplex and sequenced on a NextSeq500 sequencer (Illumina, San Diego, CA, USA). Sequence data were analyzed with SeqNext Analysis Software (JSI medical systems, Ettenheim, Germany). 30 × coverage for >95% of the coding sequence from each gene, then 30 × coverage for >98% of the total coding sequences of all genes for the total gene panel. After validation of potential pathogenic variants by Sanger sequencing, the variants were classified according to The American College of Medical Genetics and Genomics guidelines [64].

### 2.4. RACIS: Robotically-Assisted Cochlear Implant Surgery

The patient agreed to RACIS with written informed consent and the study (ClinicalTrials.gov NCT04102215) was performed with the approval of the Antwerp University Hospital ethics committee (B300201837507). The HEARO^®^ robotic system (CASCINATION AG, Bern, Switzerland) is an otological surgical robotic platform with a focus on cochlear implantation surgery (Appendix A). It integrates a set of sensors, actuators, and core functionalities to enable the surgeon to perform image-guided surgery with a robotic arm. The HEARO procedure for RACIS comprises too many detailed steps that exceed the scope of this article. Here we would like to focus on the planning of the drilling trajectory. Dedicated planning software OTOPLAN^®^ (CASCINATION AG, Bern, Switzerland) allowed the surgeon to plan a 3D reconstruction of all relevant anatomical structures and designated the target on the cochlea, and planned the most suitable trajectory to the cochlear round window. This trajectory accommodated the safety distance to the critical structure while minimizing the in and out-plane angles [63]. Figure 2 shows the surgical plan and distances to the critical structure as planning details. The preferred surgical trajectory (ST) was adjusted to the patient’s specific anatomy but also had to be parallel with the scala tympani. Since the modiolus was absent, a straight extrapolation of the ST ideally hits the lateral wall of the cochlea as laterally as possible.

Cochlear access was warranted by HEARO through the keyhole trajectory, but the surgeon required a transcanal view of the cochleostomy for insertion. Initially, a 1.8-mm tunnel was created to the middle ear cavity. In the next step, a 1 mm diamond burr was used to perform the robotic inner ear access by milling away the round window bony overhang. An insertion guide tube, consisting of 2 half-pipes, was placed in the drilled tunnel to bridge the gap in the middle ear cavity and avoid a false route of the array into aerated mastoid cells (Appendix A). The tube also acts as a protective barrier for the electrode array against the blood and debris in the tunnel. A tympanomeatal flap was created as second access for the purpose of the visualization of the electrode insertion and managing the gusher. The surgeon (VT) opted for a shorter array: Synchrony FLEX 20 PIN electrode (MED-EL, Innsbruck, Austria) to fit the available one turn of the cochlea.

## 3. Results

### 3.1. Surgery Results

Full insertion of the electrode array was performed and soft tissue with fibrin glue was used for tight packing and sealing of the cochleostomy. The system accuracy for this case was 0.13 and 0.04 mm at the level of round window and facial nerve, respectively. The intraoperative mobile cone beam CT (CBCT) XCAT XL (XORAN, Ann Arbor, MI, USA) confirmed the correct and full insertion with 417 degrees of cochlear coverage at the tip (Appendix A).

### 3.2. Molecular Genetic Analysis

A positive familial history of hearing loss was present. The proband of the family was a 71-year-old woman (I:2) who had profound SNHL with late-onset and it progressed with age. Her two sons (II:1, II:2) and her daughter’s son (III:1) presented with prelingual hearing loss (Figure 3). Next-generation sequencing of a hearing loss gene panel in her grandson showed a hemizygous c.934G > C p. (Ala31Pro) likely pathogenic variant in the *POU3F4* gene (NM_000307.4). By subsequent familial screening, her two affected sons were shown to be hemizygous for this variant. Her daughter was a heterozygous carrier. This variant has not been reported before and is located in the POU homeoboxdomain. An alanine to valine substitution at the same amino acid position has been reported previously as the cause of hearing loss [17]. Apart from this, no possible variant that could be the cause of hearing loss was observed.

### 3.3. Audiological Results

Intraoperative telemetry revealed low impedances on all electrode contacts (1.75–6.03 kΩ). The first activation of the implant speech processor took place 4 weeks after surgery. With this procedure, sound field thresholds were improved with the implant system applied to the right ear. One month after activation, pure tone averages close to 43 dB HL were achieved with the CI processor and are shown in Figure 4. Despite the language barrier, speech perception improved from 0% to 50%, measured one month after activation. Speech audiometry shows phoneme scores at 0% 45 dB SPL with insert phones.

## 4. Discussion

SNHL due to x-linked deafness (DFNX2) by pathogenic variants in the POU3F4 gene were initially concealed by a consistent air-bone gap [4,39,41]. In the past, gushers have been reported during an attempt for stapedotomy, whereas stapes reflexes usually are present in these cases. The supraliminal bone conduction thresholds can be attributed to the anomaly itself that consists of an absent separation between the IAC and the cochlea. Vibrations on the skull are conducted more easily towards the inner ear, similar to the pathophysiological mechanism in superior semicircular canal dehiscence or even the Carhart effect in otosclerosis. The natural course of hearing deterioration in DFNX2 has previously been reported by Cremers et al. [65] and Snik et.al [66], so-called age-related typical audiograms. Snik et al. [66] reported that the audiovestibular system functions better than the normal system due to congenital malformation, resulting in better bone conduction levels. The authors demonstrated in their studies that the air-bone gap in the audiogram does not have an important component of conductive hearing loss and is compatible with pure SNHL. It has been stated that the SNHL seen in DFNX2 patients progresses over time, so adequate speech and language skills can be developed at an early age. It is thought that this improvement may be due to pressure changes in cerebrospinal fluid being transmitted to the perilymph [65]. In our elderly patient, we see that hearing loss progresses with age. Similarly, in a 45-year-old female patient due to the *POU3F4* mutation, it was reported that SNHL increased with age [38]. Hearing loss in elderly patients appears to affect daily functioning, communication, social participation, and quality of life. Therefore, hearing rehabilitation with CI results in improvements in speech perception, oral communication, quality of life, and cognitive abilities [67,68]. It is thought that it can directly affect the health of the person while participating more fully in daily life activities. In a multicentric study, it has been shown that the CI applied to adults over the age of 65 significantly benefits hearing and quality of life and can facilitate the concept of healthy aging [69]. Consequently, CI surgery was thought to be the most effective solution for our patients. The cross-sectional analyses reported here actually reflects the hearing deterioration in our case, leading to CI candidacy at 71 years.

Our patient, in this case, was a 71-year-old female patient and had a late profound sensorineural hearing loss that progressed over time. Profound hearing loss can be rehabilitated with CI, although the surgery is challenging due to their inner ear anomaly. There are significant risks in surgery, such as gusher during cochleostomy and electrode misplacement into the IAC [45,58]. For this reason, surgeons have developed some applications to avoid possible risks in surgeries. It has also been reported that straight electrodes directed towards the lateral wall in these patients with IP-III anomaly may minimize the chance of entering the IAC and shorter electrodes will decrease the possibility of accidentally entering the IAC [45,53,61]. Since the exact location of the neural tissue in patients with IP-III anomaly is not known exactly, it is recommended to stimulate as much neural tissue as possible by using electrodes with full contact surfaces or electrodes with active electrodes on both surfaces in these patients [61]. Conversely, it has been stated that full band electrodes may be risky due to incorrect facial nerve stimulation [62]. Postoperative facial nerve stimulation in children with IEM is likely due to the proximity of the facial nerve and the electrode array, the abnormal course of the facial nerve, or it is caused by the electrode being placed in the IAC by mistake [70]. However, since we calculated the trajectory of the electrode in advance with robotic surgery, we used a full band electrode because we minimized the incorrect facial nerve stimulation. Besides, it was stated that CI in a severe gusher case would be the easiest way to manage if the surgeon waits for the gusher to stop before placing the electrode [53,59]. In our case, we waited for the severe gusher to pass, and we minimized possible complications using robotic surgery.

Robotic technology is thought to allow minimally invasive cochlear access, a controlled electrode placement process, and is also unaffected by human operator limitations [71]. Since the cochlear approach with robotic surgery is planned according to the anatomical structure of the patient, it reaches the cochlea by deviating at a tenth of a mm level with the keyhole trajectory. Optimizing such sub-millimetric settings requires extensive surgical skill and experience in conventional CI surgery [62]. Additionally, estimation by even experienced surgeons of angles and direction appears to be imprecise [72]. Due to such fine surgeries and different anatomical structures due to anatomical variations, it is now possible to support surgeries with robots and to receive support from simultaneous imaging methods during surgery with the rapid advancement of technology. For this, some researchers have pioneered the discovery and application of robotic CI [73,74]. It was also applied by Weber et al. in 2017, through developing an image-oriented robotic arm [71], and thereafter cochlear approach was also applied safely by Topsakal et al. by controlling the angles in the robotic surgery [63]. Robotic middle ear intervention was performed for the first time by Caversaccio [75], while inner-ear access was performed for the first time by Topsakal [63]. In addition, it was ensured that the surgeon avoided proactively damaging the facial nerve by integrating it into the robotic system during robotic drilling in the mastoid and monitoring the facial nerve [76]. Deafness cases attributed to DFNX2 will hopefully pose a less surgical challenge with technological improvements. We have demonstrated that a robotic approach requires much more correct planning rather than surgical experience. We anticipate more surgeons can counsel more patients with this type of anomalies towards CI.

Here we illustrate the usefulness of RACIS in an IP-III anomaly. In CI surgery performed on patients with IP-III anomaly, gusher was reported in all cases in many studies [34,45,50,53,62,77,78]. The choice for a cochleostomy proved efficient for gusher management through the ear canal. A round window gusher would be more difficult to even visualize in this approach. It is very difficult to estimate the shape and size of the cochleostomy during the severe gusher. Cochleostomy looks different from its actual size due to cerebrospinal fluid exit and dimensions [61]. The gusher occurred moderately probably because the diamond burr on the robot arm was not redrawn from its position. In a certain way, this functioned as a stopper, and after some time the gusher change into a heartbeat synchronous pulsating leakage. After full insertion of the electrode array, the gusher subsided and tight packing stopped all oozing. No cerebrospinal fluid-leakage was observed in follow-up of up to 9 months. As studies are showing the long-term audiological benefits of CI in patients with IP-III [56], we observed that the audiological parameters started to improve in the first month after the cochlear implant was activated in our patient.

We have demonstrated for the first time that RACIS can be applied safely in DFNX2 patients with the HEARO procedure, which provides us with these features. We argue that robotic surgery can never replace a surgeon, but it may reach a level such as an autopilot function to control the trajectory of an aircraft. Surgeons should reach out to this technology to standardize surgical outcomes to serve their patients.

## 5. Conclusions

IP-III anomaly, one of the rarest inner ear anomalies, is also the rarest among incomplete partition groups. Conventional CI is a safe procedure for those with severe-profound mixed hearing loss due to inner ear malformation associated with the POU3F4 mutation and CI can be recommended among the treatment options depending on the level of hearing loss, although there are some surgical difficulties. The present study reports on the first case of IP-III where robotically-assisted cochlear implantation surgery was performed. A technique that provided targeted minimally invasive inner ear access.

## Figures and Tables

**Figure 1 genes-12-00613-f001:**
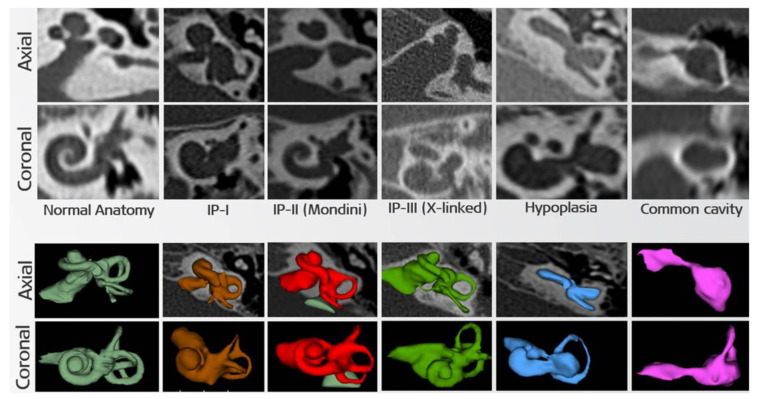
2D and 3D images of IP-III anomaly compared with normal and other inner ear anomalies in axial and coronal views (modified from Dhanasing et al. 2019 with permission) [52].

**Figure 2 genes-12-00613-f002:**
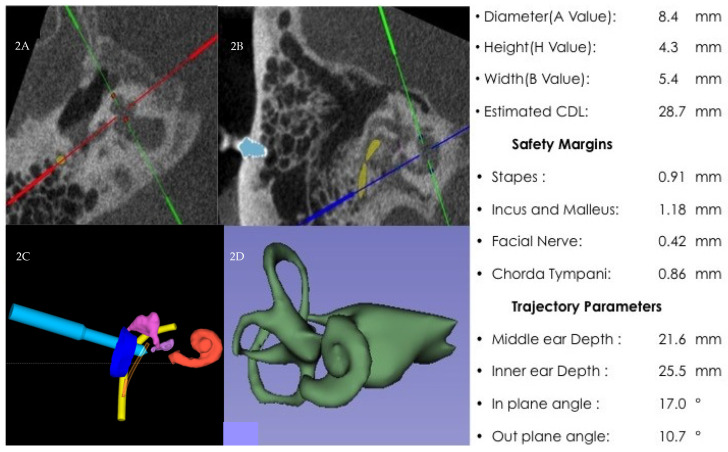
(**A**) Axial view of a mobile cone beam CT (CBCT) with green and red helpline crossing anticipated spot of modiolus, red dots depict cochlear height: 8.4 mm. (**B**) Sagittal view of CBCT with 1 fiducial colored blue and facial nerve colored yellow (colored lines cross anticipated spot modiolus). (**C**) Three-dimensional planning of the surgical trajectory. Light blue indicates the drill trajectory, purple ossicular chain, dark blue ear canal, yellow facial nerve, and orange is cochlea. (**D**) A 3D reconstruction demonstrating the IP-III anomaly: Cochlea with a short modiolus and less overall length of the interscalar septa resulting in a smaller number of turns.

**Figure 3 genes-12-00613-f003:**
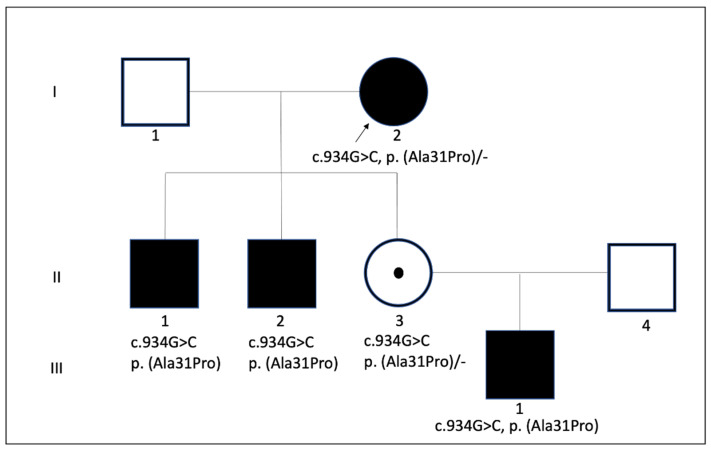
The three-generation family pedigree demonstrating segregation pattern of *POU4F3* c.934G > C p. (Ala31Pro). The arrow indicates the proband and case that underwent robotic surgery.

**Figure 4 genes-12-00613-f004:**
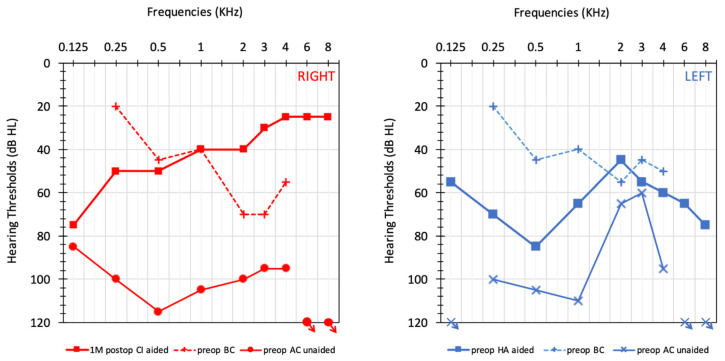
Audiometric results unaided preoperative air conduction (AC) and bone conduction (BC) thresholds can be observed in red O and blue X for right and left side, respectively. Pre-operative best-aided thresholds (patient wears hearing aid (HA) only on left side) on the left side (blue). Post-operative thresholds in the best-aided setting with CI after 1 month on the right side (red). Speech audiometry shows phoneme scores at 0% 45 dB SPL with insert phones.

**Table 1 genes-12-00613-t001:** Mutations in the *POU3F4* gene reported in literature to result in DFNX2 phenotypes.

Nucleotide Change	Amino Acide Change	Feature of Deafness	Defects on Temporal Bone CT	Perilymphatic Gusher	Location	Year	References
603–610delCAAA	p.Lys202 fs	SNHL	Yes		The Netherlands	1995	[4]
c.648–651delG	p.Arg215 fs	Mixed	Yes	Yes	The Netherlands	1995	[4]
c.895delA	p.Leu298 fs	Mixed	Yes	Yes	The Netherlands	1995	[4]
c.950T > G	p.Leu317Trp	Mixed	Yes		The Netherlands	1995	[4]
c.1000A > G	p.Lys334Glu	Mixed	Yes		The Netherlands	1995	[4]
c.862del4	p.Ser288Gln	Mixed	Yes	Yes	UK	1995	[17]
c.935C > T	p.Ala312Val	SNHL	Yes		UK	1995	[17]
del 2.6 kb, 6.5 kb					Korea	1996	[5]
del 7 kb, 4.4 kb					Korea	1996	[5]
del 8 kb		Mixed	Yes		Korea	1996	[5]
del 20 kb		Mixed	Yes		Korea	1996	[5]
del 130 kb		Mixed	Yes		Korea	1996	[5]
del 200 kb		Mixed	Yes		Korea	1996	[5]
del 220 kb		Mixed	Yes		Korea	1996	[5]
del entire gene		Mixed	Yes		Korea	1996	[5]
c.967C > G	p.Arg323Gly	Mixed		Yes	The Netherlands	1997	[10]
c.990A > T	p.Arg330Ser	SNHL	Yes	Yes	The Netherlands	1997	[10]
c.985C > G	p.Arg329Gly	Mixed	Yes	Yes	US	1997	[31]
c.689C > T	p.Thr230Ile	Mixed	Yes	Yes	US	1997	[31]
601–606delTTCAAA	p.Phe201/Lys202del	Mixed	Yes	Yes	Japan	1998	[26]
del 1200 kb		SNHL	Yes		Spain	2000	[32]
del530 kb		SNHL	Yes	Yes	US	2005	[18]
c. 683C > T	p.Ser228Leu	SNHL	Yes		US	2005	[18]
c.925 T > C	p.Ser309Pro	SNHL	Yes		China	2006	[19]
c.383delG	p.Gly128fs	SNHL	Yes		Korea	2009	[33]
c.623T > A	p.L208Ter	SNHL	Yes		Korea	2009	[33]
c.986G > C	p.Arg329Pro	Mixed	Yes		Korea	2009	[15]
c.927–929del	p.Ser310del	Mixed	Yes		Korea	2009	[15]
c.293C > A	p.Ser98Ter *	Mixed	Yes		France	2009	[28]
	p.Glu236Asp *	Normal	No		France	2009	[28]
	p.Arg282Gln *	Normal	No		France	2009	[28]
	p.Ile285Asn *	Normal			France	2009	[28]
	p.Ser288CysfsTer40 *	Normal			France	2009	[28]
	p.Ile308Asn *	Mixed	No		France	2009	[28]
	p.Ile308 IlefsTer28 *	Normal	No		France	2009	[28]
c.499 C > T	p.Arg167Ter	Mixed	Yes	Yes	Korea	2010	[34]
c.647G > A	p.Gly216 Glu	SNHL	Yes		China	2010	[20]
c.973 T > A	p.Trp325Arg	SNHL	Yes	Yes	Germany	2011	[35]
c.341G > A	p.Trp114Ter	SNHL	Yes		Pakistan		[21]
c.406C > T	p.Gln136Ter	SNHL	Yes		Pakistan		[21]
c.235C > T	p.Gln79Ter	SNHL	Yes		Israel		[36]
c.853 854del	p.Ile285Argfs43Ter				Israel		[36]
c.623 T > A	p.Leu208Ter	SNHL	Yes		Korea	2013	[8]
c.632C > T	p.Thr211Met	Mixed	Yes		Korea	2013	[8]
c.686A > G	p.Gln229Arg	SNHL	Yes		Korea	2013	[8]
c.950dupT	p.Leu317PhefsTer12	SNHL	Yes		Korea	2013	[8]
c.1069delA	p.Thr 354GlnfsTer115	SNHL	Yes		Korea	2013	[8]
c.1084 T > C	p.Ter362ArgextTer113	SNHL	Yes		Korea	2013	[8]
c.987 T > C	p.Leu308Thr	Mixed			Nigeria	2015	[22]
c.902C > T	p.Pro301Leu	Mixed			Ecuador	2015	[22]
c.772delG	p. Glu 258Argfs	SNHL	Yes		Turkey	2015	[22]
c.707A > C	p.Glu236Ala	SNHL			Turkey	2015	[22]
c.346delG	p.Ala116Profs	SNHL			Turkey	2015	[22]
c.727_728insA	p.Asn244LysfsTer26	SNHL	Yes	Yes	Japan	2015	[27]
c.79C > T	p.Gln27Ter	SNHL	Yes	Yes	Poland	2016	[37]
c.346delG	p.Ala116Profs	SNHL	Yes	Yes	Poland	2016	[37]
c.559G > T	p.Glu187Ter	SNHL	Yes		Poland	2016	[37]
c.623 T > A	p.Leu208Ter	SNHL	Yes	Yes	Poland	2016	[37]
c.650 T > A	p.Leu217Ter	SNHL	Yes	Yes	Poland	2016	[37]
c.823C > T	p.Gln275Ter	SNHL	Yes	Yes	Poland	2016	[37]
c.916C > T	p.Gln306Ter	SNHL	Yes		Poland	2016	[37]
c.971 T > A	p.Val324Asp	SNHL	Yes	Yes	Poland	2016	[37]
c.982A > G	p.Lys328Glu *	SNHL			Taiwan	2017	[38]
c.499C > T		Mixed	Yes		China	2017	[39]
c.927delCTC	p.Ser310del	Mixed	Yes		China	2018	[40]
c.669 T > A	p.Tyr223Ter	SNHL	Yes	Yes	China	2018	[40]
c.973delT	p.Trp325GlyfsTer12	Mixed	Yes		China	2018	[40]
c.975G > A	p.Trp325Ter	Mixed	Yes		Russia	2018	[23]
c.852delC	p.Ile285Serfs;Ter3	Mixed	Yes		Korean	2019	[41]
c.76C > T	p.Gln26Ter	Mixed	Yes		China	2019	[24]
c.604 A > G	p.Lys202Glu	SNHL			Vietnam	2019	[25]
c.699C > A	p.Cys233Ter	SNHL	Yes	Yes	China	2020	[42]
c.400_401insACTC	p.Gln136LfsTer58	SNHL	Yes	Yes	China	2020	[29]
c.870G > T	p.Lys290Asn	Mixed	Yes		Italy	2020	[30]
c.934G > C	p.Ala31Pro *	Mixed	Yes	Yes	Turkey	2020	Present Study

* Asterisk indicates correspond to findings in females. Empty rows indicate “unspecified” in the report.

## Data Availability

The data presented in this study are available in the article and Appendix A.

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
