# Peer review of "A New Pathogenic Variant in POU3F4 Causing Deafness Due to an Incomplete Partition of the Cochlea Paved the Way for Innovative Surgery"

_genes, 2021, doi:10.3390/genes12050613_

Round 1
Reviewer 1 Report
Please see attached file

Reviewer 2 Report
Dear Authors,
Thank you for submitting this interseting and precisely described report on robotically assisted cochelar implant surgery in case of a new pathogenic variant of POU3F4.
The article comprises relevant information,.
Besides display of a new variant of POU3F4, to date knowledge on variants in the gene are reported. A further nice aspect is the intersesting description and discussion on the phenotype of a female carrier.
Moreover, indication and conduction of robotically assisted CI surgery in case of IEM is comprehensibly argued.
Some minor adaptions would be recommended:
- is the patient herself a heterozygous carrier, like her daughter, or was the variant present in a homozygous state - clinically not expected but worth mentioning
- p2, l.46: rephrase sentence - mind "caused by"
- please provide information whether informations on variants displayed in table 1 may be found in public databases, e.g.,clinvar
- concerning description of Gusher management during surgery: I would add this information in section 2.4; there is information on page 15 l. 296-298 and lines 321 ff., however it becomes not clear, why robotic surgery has an advantage in gusher management; waiting for the gusher to relief is not an advantage of robotic surgery; moreover, may there be a disadvantage in application of soft tissue due to a smaller surgical field?
Small corrections concerning typo or sentence structure:
p.9, l.131: missing word after correct
p.14, l.258: parts missing at beginning of sentence
p.14, l.274: missing comma after case
p.15, l.281: check for spacing after IAC
p.15, ll. 216-320: please repharse sentences, check for grammar
Reviewer 3 Report
The manuscript provides a novel an interesting area. Indeed, the manuscript also presents a novel finding of a pthogenic variant in POU3F4, however, without any validation in animal models. In my opinion, this is not necessary considering the occurance of this variance in a set of relatives. The methods and results are presented in a high quality manner. The introduction is too long and lacks focus. The majority of the discussion reads like a review article on the different POU3F4 variants causing hearing loss. Based on the title, the reader expects a novel variant and a novel way of innovative surgery. The authors should decide whether the focus is on the new variant or the robotic surgery and then prepare the reader in a concise manner in the introduction.
The authors should indicate in the title the design of the study, which is a case report.
The discussion is also too long and sometimes lacks focus. According to the introduction, the discussion section would improve by ommiting redundant sentences (e.g., line 275-277) and by foccussing on the overall aim of the study.
